# EduARdo—Unity Components for Augmented Reality Environments †

**Ilias Logothetis *, Myron Sfyrakis and Nikolaos Vidakis**

Department of Electrical & Computer Engineering, Hellenic Mediterranean University,
71004 Heraklion, Greece; nv@hmu.gr (N.V.)
* Correspondence: iliaslog@hmu.gr; Tel.: +30-2810-379117
† This article is a revised and expanded version of a paper entitled Hand Interaction Toolset for Augmented Reality Environments, which was presented at XR Salento Lecce (Italy), 6–8 July 2022.

**Abstract:** Contemporary software applications have shifted focus from 2D representations to 3D. Augmented and Virtual Reality (AR/VR) are two technologies that have captured the industry's interest as they show great potential in many areas. This paper proposes a system that allows developers to create applications in AR and VR with a simple visual process, while also enabling all the powerful features provided by the Unity 3D game engine. The current system comprises two tools, one for the interaction and one for the behavioral configuration of 3D objects within the environment. Participants from different disciplines with a software-engineering background were asked to participate in the evaluation of the system. They were called to complete two tasks using their mobile phones and then answer a usability questionnaire to reflect on their experience using the system. The results (a) showed that the system is easy to use but still lacks some features, (b) provided insights on what educators seek from digital tools to assist them in the classroom, and (c) that educators often request a more whimsical UI as they want to use the system together with the learners.

**Keywords:** extended reality; augmented reality; user interaction





## 1. Introduction

Extended Reality (XR) is a term widely used these days. It has emerged as an umbrella term to describe all technologies that blend the real and virtual worlds [1], with the two best-known examples being augmented reality (AR) and virtual reality (VR). These technologies allow users to display virtual content in the real world (AR) [2], blend the real and virtual worlds (Mixed Reality—MR) [3], or fully immerse themselves in a virtual world (VR) [4]. As these technologies continue to evolve, the need for improved interaction in their corresponding applications becomes increasingly important.

Traditional means of interaction, such as the use of a mouse and keyboard, a touch-screen, or a controller, may be convenient in some cases but limit the immersive experience and realism expected by users [5]. In addition, the requirements of external interaction devices necessitate time and effort for familiarization and can shift the focus from the content to the operation of such devices [6]. To increase the level of immersion, researchers are exploring more natural and realistic ways to interact within these environments [7–10].

To this end, various hardware devices capable of tracking hand movements have been developed, such as Microsoft's Kinect [11], LeapMotion [12], and wearables such as gloves [13]. These devices provide high accuracy, but they are either limited to a specific location or require specialized equipment to set up. A modern approach is to attempt to detect and track hands using a camera. This approach offers the advantage that almost every smart device has a camera. While this method does not require any external hardware to operate, the hands must be in the camera's field of view. Modern VR headsets such

as Meta Quest 2 include hand-tracking functions based on the camera use of the mobile device. Depending on the application requirements, one technique may be preferred over another, as each has advantages and disadvantages.

The estimation of the exact position of a hand is also possible through techniques other than hand detection and tracking, allowing for the execution of high-precision actions. These methods tend to be computationally intense, using a variety of algorithms to predict and reconstruct the exact position of a hand. The recognition of specific hand gestures is a technique that is not as computationally intense. When high precision is not required, the latter method is preferred. If hardware resources are limited, the second method is also preferred. Both can provide similar functionalities, favoring a natural type of interaction. Such functionalities include grabbing, moving, and detecting collisions with virtual objects. These are simple tasks that any application in a three-dimensional environment integrates with traditional interaction techniques.

As XR technology evolves, the development and integration of natural interaction methods into these environments becomes increasingly important. The existing technologies, both software and hardware, are rapidly advancing, requiring developers to constantly learn new tools, and forbidding them from reusing existing components [14]. In addition, companies oblige developers to use specific tools. The lack of frameworks and libraries with which to facilitate this process can discourage developers from engaging with this technology.

The proposed system addresses this need by providing a collection of tools independent of the underlying hand-tracking algorithms, making it easier to create engaging and immersive experiences. In addition, the separation of the hand-tracking algorithms, in a specific module, provides unlimited development options. One option is to access the hand-tracking algorithms from an external web service that is hosted, and a second option is to replace it with external hardware. Both options reduce the processing load on mobile devices, thus facilitating the use of XR technology on cheap devices. Moreover, this separation allows for an easier transition to a different hand-tracking algorithm if needed.

This study provides an agile toolset that developers may use to incorporate interactions with hands and the configuration of the movement behavior of objects placed in the scene. Although the tool targets the development of AR environments and scenarios, it can also be used in any other XR context.

The system aims to support developers and users without a software-engineering background. The primary focus group is educators who want to incorporate technology into their teaching using a platform that can meet the demands of a user from a 3D development platform. For this reason, the study includes a system usability evaluation and interviews with educators. In software development, the reception of end user feedback is always important for increasing the usefulness and acceptance of a system.

The rest of the paper is organized as follows: Section 2 presents the state of the art, Section 3 analyzes the proposed system, Section 4 describes the experimental setup and system evaluation, Section 5 presents the study's results, and Section 6 concludes with a discussion of the results and the system. Finally, Section 7 summarizes the study's findings and offers suggestions for future research.

## 2. Related Work

Freehand interaction is critical in AR environments, which is why there is voluminous research on this topic, ranging from the study of hand-tracking techniques to better ways to handle interactions. With regard to hand tracking, there are many different approaches and methodologies, some of which include the tracking of hands using a camera [14] or wireless systems [15]. Many devices geared toward AR and VR have adopted a hand-tracking technique to offer such functionality to end users. Some devices that include hand-tracking functionality or are even built specifically for this task include LeapMotion, Kinect, HoloLens2, and Meta Quest 2.

Most developers or creators seeking to develop applications in AR require more than just the means to track a hand. There is a need for high-level tools that can enable the functionality of users' hands within the applications. Tools and systems are emerging for such tasks, including GesturAR [16], which enables users to configure their gestures and actions with the objects of the scene. Furthermore, this capacity allows users to define behaviors such as transformations of the objects within a scene. Another tool is the Hand Interfaces system [17], which lets users use their hands as tools. Likewise, virtualGrasp [18] uses gestures performed during interaction with physical objects to retrieve the corresponding digital objects inside a virtual scene. MagicalHands [19] is another tool that employs gestures to create animations in VR environments. While these tools provide a captivating way in which to define new gestures and animations, their design does not allow them to act as tools for creating AR applications. Such tools can behave as applications that enable the configuration of some mini-scenes or behaviors within an application.

They require time for setup and do not always cover the case of new content creation. In addition, while this type of interaction feels natural to young people engaging in a virtual environment, older people that are unfamiliar with interacting with technology might expect more traditional means with which to configure their scenes (such as checkboxes and drop-down menus) [20]. This is the case with EduARdo, which aims to help educators who are unfamiliar with technology.

Despite the importance of freehand interaction in AR, other means include voice-based, gaze-based, location-based, or even tactile interaction, which have proven to be well-suited to specific situations. Combined interaction techniques were proposed in [21] to assist the user. For these types of interaction, the same need applies, i.e., developers and creators require higher-level tools that will allow them to configure the desired behaviors and functionality within a few simple steps. EduARdo aims to address this need by providing a simple and intuitive way in which to configure such functionality for every possible use.

Interaction is a large component, but there are many more crucial components of an application. For instance, there is a need to place objects within a scene and configure their actions. Another important component is the User Interface (UI), such as menus and text fields that present information. As technology advances, it departs from 2D content—which tends to become obsolete—and everything moves toward 3D visualizations, for which there is a more complex procedure of placing content within a scene and interacting with it. Even on modern websites, there are 3D objects, or, in some cases, the whole site is a 3D environment. In AR and VR, which are terms mostly used with regard to games, a 3D environment is the norm. One study [22] reported the lack of tools for application development in AR and the necessity of creating standards for the development process.

One System developed to create AR applications is ZeusAR [23], which was built using Javascript and the ThreeJS library. This tool provides a wizard that users configure to create AR serious games. A potential drawback of this tool is related to the technologies used, as they currently offer limited functionality in AR environments. This imposes limitations when a user requires a more complex application that the tool cannot support. Additionally, ZeusAR presents games in AR that have already been created with another creation tool; as a result, the creation of new content cannot ensue. EduARdo addresses these issues via its implementation in the Unity 3D game engine [24], which supports every AR functionality available and can exploit low-level APIs to configure extra functionality if required.

AR-Maze [25] is an educational tool that allows children to create their own scenes using wooden cubes with a texture. Children can arrange these cubes in the real world and create mazes in which the AR application can project virtual content. The game was developed using Unity 3D and the Qualcomm Vuforia platform for AR support. A similar framework was proposed in [26] for augmenting physical boards with pawns and other objects. Interactive Educational Content Based on Augmented Reality and 3D Visualization [27] was proposed as a tool for creating educational content using AR that

was designed for secondary education. ComposAR [28] focuses on associating virtual content with real objects and defining interactions for them.

SMADICT [29] is a framework in which teachers and students can participate in the design process. ScoolAR [30] is a system that allows teachers to create AR and VR experiences without requiring programming skills, but whose provided functionality is limited, only allowing images to be uploaded and tagged with text. The BIM-VR [31] framework proposes another approach for 360-degree videos in which a user can tag structures and items when pointing at them using a gaze interaction method. A general-purpose framework that facilitates the development of applications for Unreal Engine was suggested in [32]; however, it was only intended for VR. GLUEPS-AR [33] is a system that can include different tools for creating AR applications.

Following the previous approaches, the system proposed herein includes tools for configuring core parts of AR applications (content placement and interaction). The included tools do not share dependencies so that changes made to one do not affect the others. Visual editors assist in the configuration process, as this is the method expected by users. In addition to the proposed system, this study provides insights into what educators require from digital tools and how they intend to use them.

## 3. Materials and Methods

Based on the problem description given above and the state of the art of research, the current authors propose a system termed Education Augmented Reality Do, "EduARdo", whose architecture system is displayed in Figure 1. EduARdo is a toolset in Unity that was developed to facilitate the creation of AR applications and is mainly intended for non-experienced developers such as educators. The main idea of the system is to exploit independent tools—each for a required feature in AR applications and games—that can collaborate with each other to deliver a more comprehensive outcome. EduARdo is a system with which non-software engineers can develop AR applications using a desktop computer or a laptop with the specifications required by the Unity 3D game engine. The resulting application requires a mobile phone that can run ARCore in the case of Android devices or Arkit in the case of iPhones.

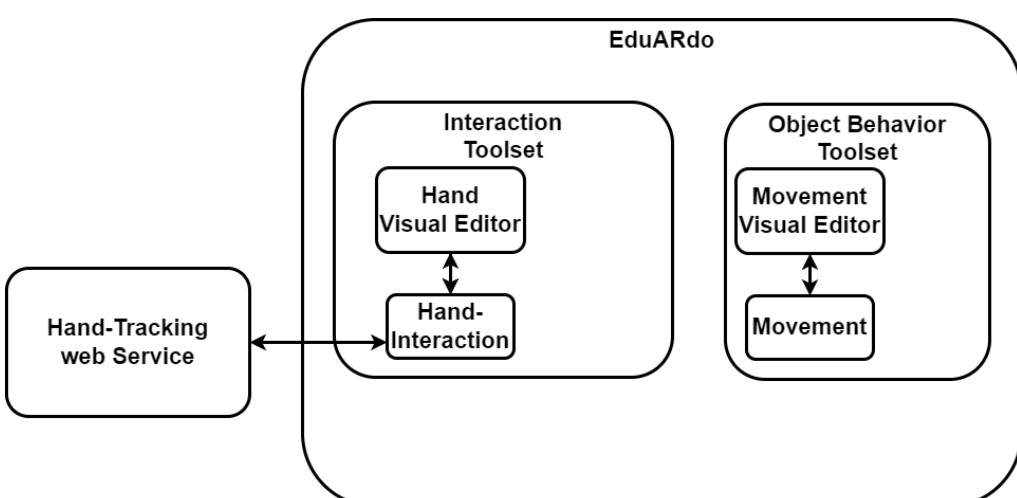

**Figure 1.** EduARdo Architecture.

In this first version of EduARdo, the authors have developed two tools. One is the interaction tool, which includes a hand interaction toolset [34]. The second tool is the object behavior tool, which aims to configure the behavior of the objects within the scene. This behavior configures the movement of objects by selecting the desired movement behavior (i.e., orbit) and then configuring the specific values of the behavior. Further description of the two tools is provided in the following sub-sections.

*3.1. Hand Interaction Toolset*

Building off of previous work [34], hand recognition was integrated in Unity (shown in Figure 2). The last version avoided this as the processing power of mobile devices was unable to handle these computations. Additionally, the early hand-tracking models required more processing power than the current models. Contemporary devices can support hand tracking and AR tracking while displaying demanding graphics. Another extension of hand tracking is the ability to use an already-hosted version of the hand-tracking module. This way, users can select the desired preference from among three options for the hand-tracking module: hosted server, local host (self-host), or integrated host.

**Figure 2.** Hand interaction toolset architecture.

Furthermore, the authors addressed an issue with the previous version of the toolset that was reported by participants unfamiliar with Unity's workflows or with little programming experience. A UI is available in this version to support such users, in which a user can select the desired configurations for their hands depending on the scene they wish to create. Figure 3 shows a breakdown of the primary components of the hand interaction toolset, as it has been reformed following the reception of user feedback, and captures the future extensions of the system. These components are described below.

- **Gesture** is a component that describes a hand gesture and provides the ability to check if a certain gesture is currently being performed. Each gesture implementation must implement the IGesture Interface to ensure that every gesture has the same behavior.
- **Fingers**—a component that describes which of the retrieved hand points correspond to each finger. By using this component, all the points of a specific finger can be collectively retrieved as a list, wherein the item in the first place of the list is the base finger point and the last item is the fingertip point.
- **GestureType** is an enumerated value that defines a type of gesture. The gesture types defined in this enumerated value are the types assigned to each gesture component.
- **GestureManager** is a component that manages all the gestures. Using this class facilitates access to the gesture classes. Additionally, it can check when a specific gesture is completed and return the object currently being interacted with.
- **HandAction** is an abstract class created to act as a guideline for the future implementation of action classes. These action classes are the result of successful interaction with an object. Sample actions include movement and rotation. Developers can implement more action classes depending on their needs.
- **SocketClient** is a simple client responsible for sending the images from the device camera to the hand-tracking module and receiving the recognized hand points.
- **ARFrameCapture** is responsible for capturing the image from the camera of the device employed. The next step is to transform the image into a form that can be sent to and processed by the hand-tracking module.

- **ActionHand** corresponds to the functionality of the virtual hand. The logic of retrieving how and when the hand can interact with an object is defined here.
- **VisualHand** is the visualization of the retrieved hand points from the hand-tracking module. This component is responsible for placing the virtual hand on top of the user's hand in the device viewport.
- **Selector** is a component responsible for containing the mechanism that will check if an interaction between the hand and an object is triggered.
- **SelectionAction** is a component that visually notifies the user about interactions with objects.
- **RayCastProvider** is a component providing the required raycaster for the corresponding task.
- **Integrated hand tracking** is a component that uses Barracuda [35] to utilize the Mediapipe [36] library for hand-tracking inside the Unity 3D game engine. This component is a modification of the open source project available on GitHub that was developed to suit the needs of this tool.

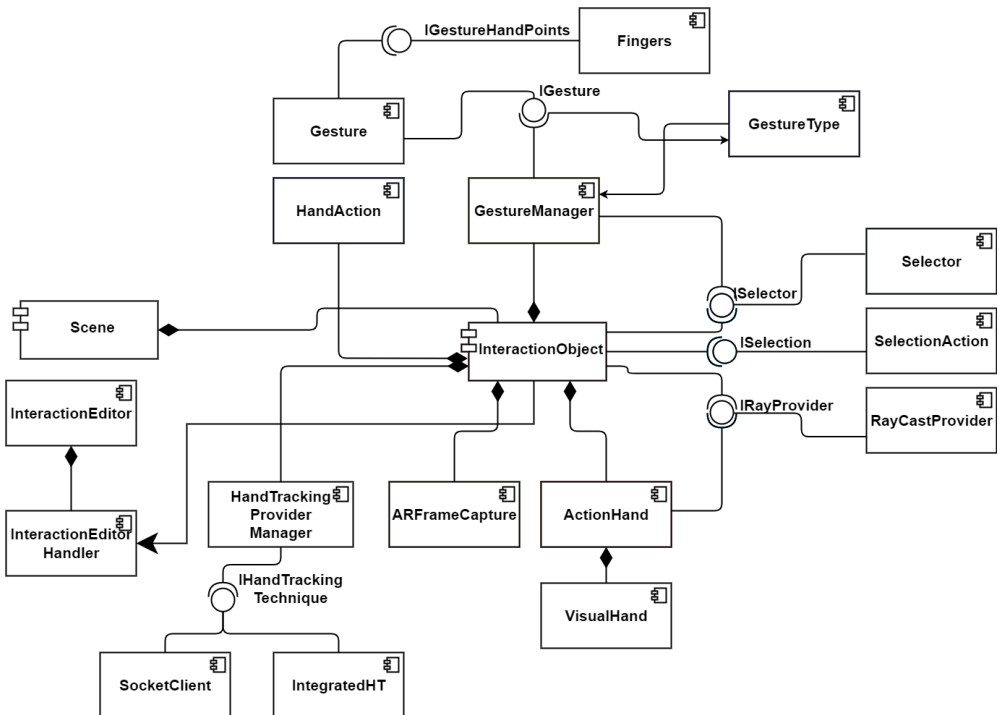

**Figure 3.** Hand Interaction Toolset Components.

The hand-tracking module was implemented in python because of the many machine-learning libraries that are implemented and maintained in this programming language. An additional factor contributing to this selection is the large community of developers that use this language for such tasks.

The latest implementation of the tool does not oblige the users to self-host this module, as the authors hosted this for the user's convenience. If users desire to self-host this module or use another hand-tracking algorithm, they can still make changes and host the module as they wish. A further alteration in this module is the hand-tracking algorithm that was switched to Mediapipe hand tracking, as it requires less computational power while still providing good tracking results. Another reason for the selection of Mediapipe is the frequent updates it offers, as the previous model is no longer maintained. The components of this module are shown in Figure 4 and are described as follows:

- **Socker Server** is the module's server. It waits for devices using the Unity toolset to connect and receive messages.

- **Image Process** handles the processing of the image received from the devices. It shapes the decoded image in the correct form for the hand-tracking algorithms.
- **Message Receiver** is a function responsible for receiving and decoding the messages sent from the devices.
- **Hand-Tracking Algorithm** is the algorithm used to recognize and track the hand in images. This algorithm is in an isolated component whose replacement is an easy task.

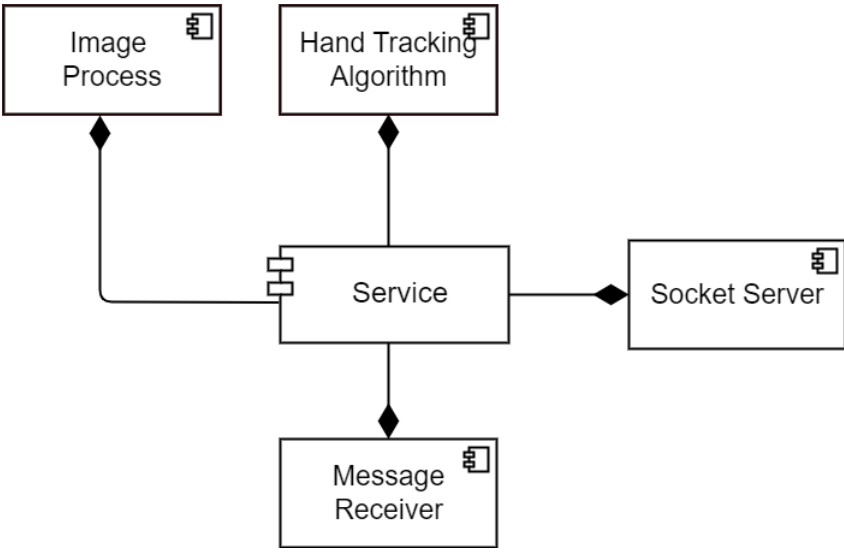

**Figure 4.** External hand-tracking module's components.

### 3.2. Object Movement Behavior Configuration

The Object Movement Behavior tool contains—as the name suggests—movement behaviors that an object can perform. Such behaviors are paths that an object must navigate by moving to specified points, performing rotations and orbits, bouncing, jumping, and more. The responsibility of this tool is to configure and store such behaviors through a visual editor designed with the UI toolkit and UI Builder in the Unity 3D game engine. This tool allows users to select the object (prefab) for which they wish to add a movement behavior; then, from the menu, they can adjust the values as they see fit to complete their use case. Figure 5 shows the abstract architecture of this tool, describing the configuration process until reaching the result, which is the selected object with the desired behavior attached to it as a Unity Component. A breakdown of the components of this module can be found in Figure 6.

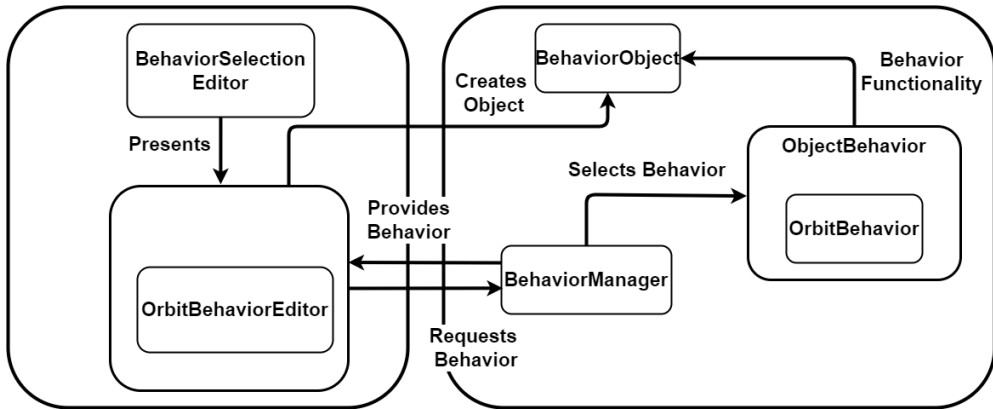

**Figure 5.** Object movement behavior tool's architecture.

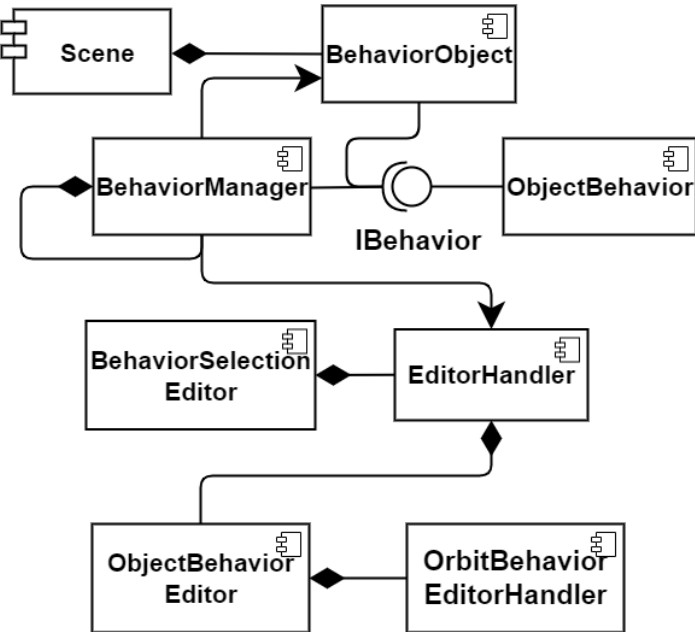

**Figure 6.** Object movement behavior tool's components.

- **ObjectMovementBehavior** is the component that holds the logic of each behavior. This component is attached to the BehaviorObject in the creation phase and contains the methods of this component that are used during run time so that the object can perform the requested behavior.
- **BehaviorManager** is a component that assists in the selection and instantiation of the correct behavior both in the application creation phase and during run time when the objects are instantiating in the scene.
- **BahaviorObject** is a prefab imported into Unity 3D game engine by the user and is the object that behavior is attached to.
- **BehaviorSelectionEditor** is a basic selector of predefined behaviors that the user can select to configure with the desired object.
- **OrbitBehaviorEditor** is a component that holds the configuration options concerning the orbit behavior that can be added to an object.

## 4. Experiment

This study aims to first measure the usability of the system and then understand what Greek educators seek from AR digital tools to assist them in the classroom. The first part of this study—the usability evaluation—is a repetition of the analysis presented in [34] with the exception that it accounts for the issues reported by the participants. To evaluate the system's usability, the participants are asked to complete tasks that will result in a simulation of the moon's orbit around earth. The application should display information about the earth or moon when a user points at them with their index finger.

The second part entails interviewing educators from Greece to obtain insights on what features they require from such systems and how they might use them. The interviews not only attempt to capture educators' perspectives on the system but on the Unity environment as well.

### 4.1. System Usability Evaluation

4.1.1. Participants

In total, 18 subjects participated in the usability study. The participants were undergraduate, graduate, and postgraduate students from Hellenic Mediterranean University, Crete. All participants had knowledge of programming, software engineering, or computer science.

4.1.2. Experiment Methodology

Potential participants received a form asking them to participate in the study. After the participants had read and signed the informed consent section, the form proceeded with a questionnaire containing basic questions intended to estimate the extent of the participants' software backgrounds. The following section provides a URL that can be accessed to download the system packages. The same URL provides a document with instructions about the tool and how to use it. As the participants were asked to use the Unity 3D game engine, the document includes download and installation instructions. Additional instructions for building the resulting application are also included in the same document. Furthermore, the same URL contains a file with the tasks that the participants were asked to complete. After the participants completed the tasks, they were requested to fill out the SUS [37] questionnaire. Lastly, an open text was made available for the users to provide their overall comments, suggestions, and misconceptions they had about the tool.

4.1.3. Tasks

The participants were asked to carry out two main tasks, which were divided into subtasks. After importing the packages into Unity 3D game engine, the first task asked the participants to use the interaction tool editor to create an empty object using the hand interaction function with the default values. The second task was to use the object movement function to create two objects. The first object should rotate around itself, while the second object should rotate around the first object. The document with the tasks provided sample numbers for the speed and orbit length. Figure 7 shows a sample of the editors the participants used.

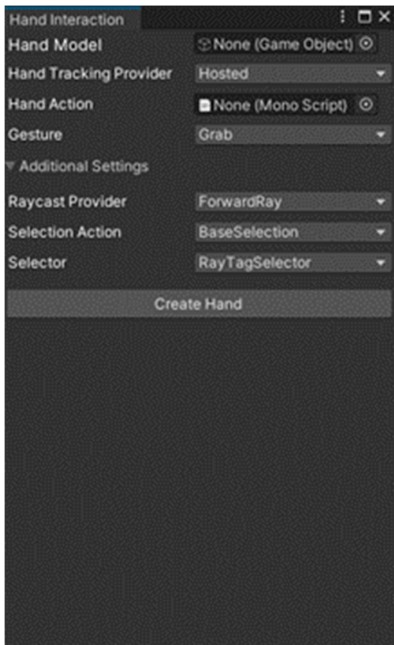
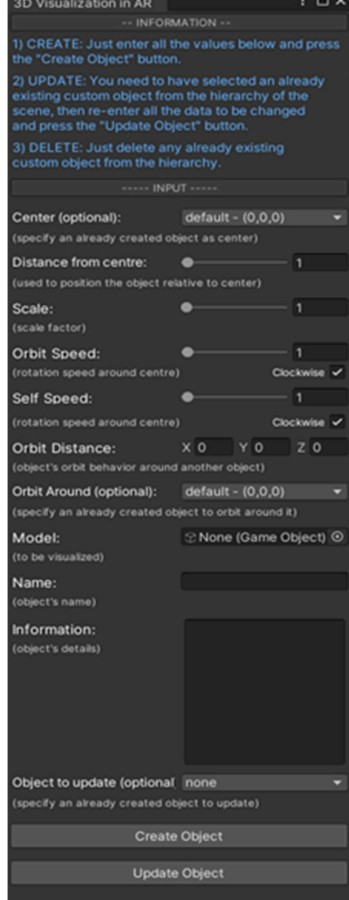

**Figure 7.** Proposed system's editors.

4.1.4. Evaluation

The SUS questionnaire was used to evaluate the tool as it is a validated method for measuring the usability of a system. The SUS questionnaire contains ten questions on a Likert scale ranging from one to five. The advantage of this method is that it can provide reliable results with a small sample size. To interpret the results, SUS provides a guide for calculating the overall score and the usability information each question contributes. According to its scale, the average score on the SUS questionnaire is 68 points.

*4.2. System Acceptance by Educators*

4.2.1. Participants

Five Greek educators without a software-engineering background and teaching in mixed educational grades participated in a semi-structured interview. The educators who participated in the study collaborated with the authors and agreed to participate to provide insight into educators' expectations of tools for creating AR learning materials and how they feel about software creation tools such as Unity. Their comments are valuable as the system mainly targets educators as their primary users.

4.2.2. Interview Methodology

For the interviews, the educators were asked to participate in a session where the authors presented the workflow in Unity and allowed the educators to experiment with the provided options while asking them their opinions of each step of the workflow until the development anddistribution of the application.

**5. Results**

All participants (i.e., educators and participants in the system usability evaluation) indicated the importance of good documentation and a good guide, which they stated would encourage them to spend time learning the system. The participants preferred a tutorial on Unity and the system in video format, while a PDF format was also requested as a supplementary file.

*5.1. System Evaluation (SUS)*

A total of 18 participants completed the SUS questionnaire. The calculation of the SUS scores was conducted according to the method described in [38]. An average score of 86.80 (in comparison, the average score in SUS is 68) was obtained, which translates to a score of A+. Figure 8 outlines the answers to each question in greater detail. In the figure, red denotes the users with a very low level of experience in Unity, while purple indicates the users who consider themselves experts. To better understand the scores, the participants were grouped based on their Unity experience. Figure 9 shows the calculated average scores of each question based on the Unity experience of the users. Means, standard deviations, and coefficients of variation for each question were calculated for further analysis of the data, see Table 1.

**Table 1.** SUS questionnaire with resulting average scores.

| | Question | Average Score | Standard Deviation | Coefficient of Variation |
|---|---|---|---|---|
| 1. | I think that I would like to use this system frequently. | 3.94 | 1.020 | 0.111 |
| 2. | I did not find the system unnecessarily complex. | 3.72 | 0.447 | 0.262 |
| 3. | I thought the system was easy to use. | 4.44 | 0.761 | 0.350 |
| 4. | I think that I would not need the support of a technical person to be able to use this system. | 3.22 | 1.030 | 0.171 |

**Table 1.** *Cont.*

| | Question | Average Score | Standard Deviation | Coefficient of Variation |
|---|---|---|---|---|
| 5. | I found that the various functions in this system were well integrated. | 4.50 | 0.600 | 0.579 |
| 6. | I did not think there was too much inconsistency in this system. | 3.61 | 0.590 | 0.133 |
| 7. | I would imagine that most people would learn to use this system very quickly. | 4.50 | 0.500 | 0.425 |
| 8. | I did not find the system very cumbersome to use. | 3.72 | 0.558 | 0.111 |
| 9. | I felt very confident using the system. | 4.44 | 0.684 | 0.436 |
| 10. | I did not need to learn a lot of things before I could start using this system. | 3.61 | 0.487 | 0.154 |

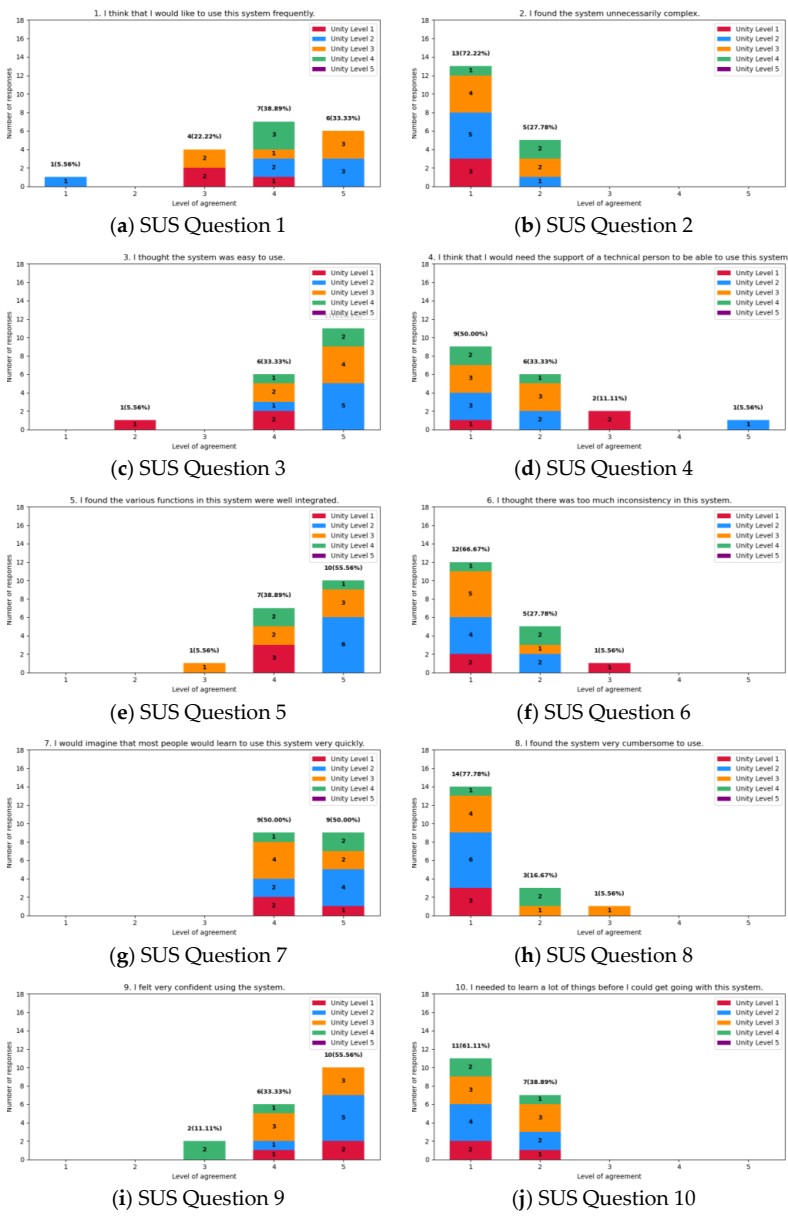

**Figure 8.** SUS answers per question.

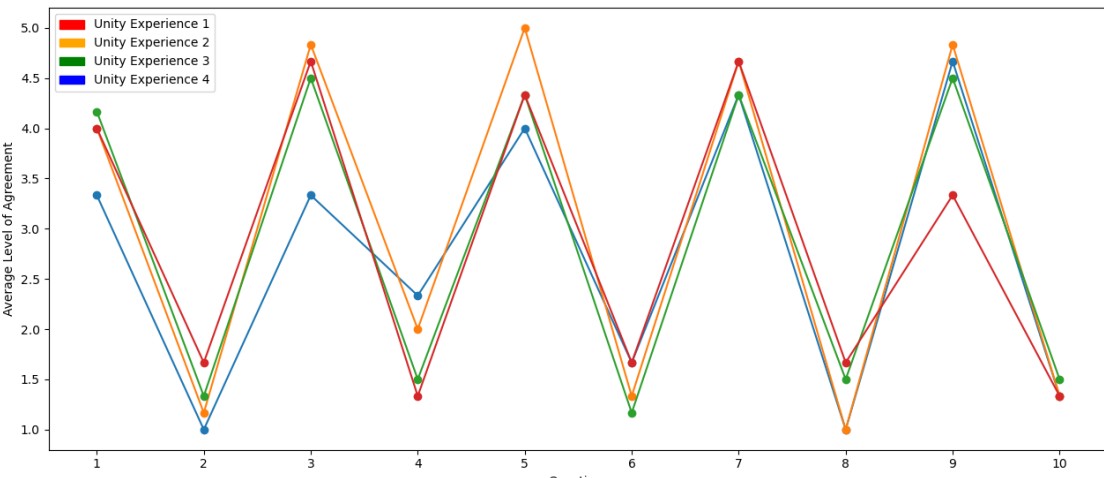

**Figure 9.** Average scores on questions per Unity experience level.

The participants in this study outlined that they require live feedback, use Unity's tooltips for additional information, or employ tooltips as the only way to present information about the editor's fields. The users requested improvements in the UI in cases of a missing or incorrect value. One participant asked for a measurement unit on the speed and scale options.

### 5.2. Insights from Educators

Thematic analysis was employed to process the qualitative results. The data under consideration emanated from interviews conducted with educators based in Greece. Figure 10 outlines the results of the study. The discussion focused on three key themes: (a) the Unity 3D game engine as a development environment, (b) the proposed system as a tool, and (c) the expectations of the participants concerning the system. According to the findings, the users suggested that the time required to learn the tool should be short, allowing them to begin creating content within a few hours.

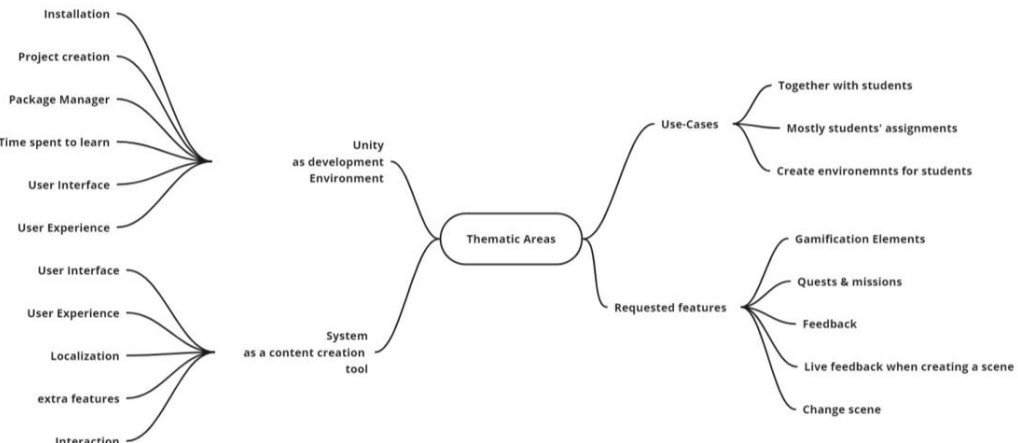

**Figure 10.** Mind map of the thematic areas retrieved from interviews.

### 5.2.1. Unity Development Environment

The Unity development environment received unfavorable feedback from the participants, who described it as unappealing and not user-friendly. One participant commented that "the environment is uncool". While the installation and project creation processes were straightforward, the participants preferred to have access to an already-created project that requires minimal or no configuration. The package manager was not favored, with one participant stating, "I could use Unity's Package Manager to import packages bought from

the asset store, but I would prefer the system to offer me many objects to start with and use the Package Manager only in rare situations".

### 5.2.2. EduARdo

The participants requested a measurement system for speed and scale and more hidden menus to improve the readability of the system's features, and they expressed concern regarding the use of the English language, especially if specialized terms are introduced or too much text is displayed.

### 5.2.3. Requested Features

The participants were asked what features they require from a software tool for AR content creation; they replied that live feedback on the configurations made using the system is the most important feature. One participant's response was as follows: "I like that iron-man interaction with virtual objects using my hands, but it would be harder to learn and use it". While the mentioned feature was visually outstanding, the development of accurate positioning or animations was deemed nearly impossible to accomplish, and the participants noted that this feature would only be comfortable for use with smart glasses rather than a smartphone. Gamification elements, quests, and missions are also critical for educators, so the ability to create such elements should be included in an educational tool. The participants also noted the importance of collaboration with other educators to create content. They also requested the ability to provide feedback for users when completing tasks or levels; change scenes; and change the color of the objects in a scene.

### 5.2.4. Scenarios of Usage

When asked how they intended to use the system, the participants expressed a desire to use it cooperatively with their students, which they claimed would encourage students to create their environments and scenes while the educators act as assistants and motivators. Some participants defined scenarios, describing pre-configured scenes that students would engage in. One participant said, "I would add objects in a scene, add text and voice recordings, and have students play in this scene, interacting with the placed objects".

## 6. Discussion

### 6.1. SUS Results

The usability of the system, as determined by the SUS evaluation, is above average. A total of 18 participants who completed the SUS questionnaire gave an average rating of 86.80. Questions 4 and 10 concerned the provision of feedback on the learnability of the system. These two questions received the lowest average scores of 3.22 and 3.61, respectively. This indicates that the users felt they needed time to learn the system before using it. The remaining eight questions were related to the system's ease of use. Questions 5 and 7 had the highest scores, with an average value of 4.50. The scores for questions 1, 3, and 9, confirm that the inclusion of an editor in Unity made the system much easier to use. Questions 5, 6, and 9 generally describe the system's architectural design as well-structured. However, question 6 had the lowest score among all the usability questions, resulting in an average score of 3.61. This could be due to a lack of feedback when an error occurred along with the live preview function, which were both features requested by most participants.

Questions 4 and 1 presented the highest standard deviations, namely, 1.030 and 1.020, respectively. Figure 8a,d indicate that the reason for the high standard deviation is that one user provided a different rating than the rest of the group. This user provided a self-evaluated score of 2 in the question about the Unity experience level, which translates to an inexperienced user.

### 6.2. Qualitative Results

The qualitative results suggest that the educators felt intimidated by Unity's user interface and felt that it was, as described, an uncool work environment. This conclusion

stems mainly from the colors Unity has chosen for its UI rather than the menu toolbar and project configuration process. Hence the need for an option to change the colors of the system editors to at least provide a more inviting environment. Educators do not want to spend time searching for or configuring Unity's options, so a ready-to-work, downloadable project is highly desirable. To meet this expectation, the system should include many 3D objects so that educators can build their environments without the need to search the internet or the asset store to find them. It is critical that the system does not contain long sections of text, as educators noted that in a foreign language, they can be confusing and discourage users from using a tool. They also showed interest in alternative interaction options, but this option was mainly for experimentation and entertainment, as they also favored the standard interaction methods.

The educators indicated that the main use case of a digital tool for education is students' use of the tool toward course material, with teachers acting as assistants. The product will be a scene, environment, or application that generally captures the learners' perspectives on the concept introduced by the teacher. To this end, the design of editors should be easy for learners to understand. This design should accommodate the aforementioned long-text problem with respect to foreign languages. The educators asked for the ability to configure gamification elements, quests, missions, and, to a lesser extent, scene-switching features. Gamification elements such as badges or points are important and can act as feedback within a scene. They have been proven to increase learners' motivation [39]. Quests and missions are crucial as they serve as representations of educational goals. Based on the interviews, it was concluded that most of the educators (a) were unwilling to learn new technologies and were usually intimidated by the complexity of digital tools; (b) required a platform that can be used to create content in many scenarios; (c) believed that the current version of EduARdo lacks the visuals required to make it appealing to educators; (d) did not request features capturing usage data and user behavior; (e) did not request the addition of a method for setting and adjusting difficulties. The authors of this paper believed that these features are the most requested by educators.

As EduARdo incorporates more tools, care must be taken to ensure that these tools can be easily accessed and used by users.

*6.3. Overview*

The most demanded items by educators and participants in the SUS questionnaire were good documentation and tutorials with examples. The requested tutorials should be videos wherein someone presents a process step by step while the user follows clear instructions with a known outcome. A live preview during the creation process is another crucial and highly anticipated feature, as it can facilitate the development process by presenting and modifying the content in the environment. The demand for these items proves that it is necessary for creators to constantly receive visual feedback on what they are making. The visual editors of EduARdo are simple to use but they require enhancements to make them amiable to users that are unaccustomed to Unity.

## 7. Conclusions and Future Work

In this study, a system, developed using the Unity 3D game engine, intended to facilitate content development in AR—and by extension XR—was proposed. The current version of the system consists of two tools, one for creating interactions and one to assign movement behaviors to the 3D objects within a scene. This version of the interaction toolset contains a hosted version of the hand-tracking service and an integrated hand-tracking module within Unity. Providing ready-to-use options in the system assists inexperienced users and saves time for experienced users. To evaluate the usability of the system, the participants were asked to answer a SUS questionnaire and rate the most favorable features with a score of A+. The results show that the system is easy to use but still lacks key features requested by the participants. An example of such a feature is a live preview of what the users are building. Additionally, the interviews with educators provided insight into the

affordances that educators desire from digital tools to assist them in classroom or course preparation. The educators heavily weighted the importance of good documentation and tutorials in video format, as this would motivate them to start working with the system. Another insight from the educators involves the UI of the system, for which they requested a more whimsical UI since they wanted to use the system together with the learners. The study showed that EduARdo provides tools that can assist the development of AR applications but still lacks many features to be considered a favorable option for educators. The system's architecture is well structured, and by following this approach, more tools can be included to create a complete system that educators can use. The next step is to implement tools for object instantiation and configuration tools. Developers should create good documentation and tutorials of the already-created tools and make them available in many languages. The ability to change the colors of the editors should also be a priority according to the educators' suggestions. Finally, after addressing these issues, this study needs to be repeated with educators only to measure the usability and acceptance of the system among this demographic.

**Author Contributions:** Conceptualization, I.L.; methodology, I.L. and N.V.; software, I.L. and M.S.; validation, I.L. and N.V.; formal analysis, I.L. and N.V.; data curation, I.L.; writing—original draft preparation, I.L. and N.V.; writing—review and editing, N.V.; visualization, I.L. and N.V. All authors have read and agreed to the published version of the manuscript.

**Funding:** This publication is financed by the Project "Strengthening and optimizing the operation of MODY services and academic and research units of the Hellenic Mediterranean University", funded by the Public Investment Program of the Greek Ministry of Education and Religious Affairs.

**Data Availability Statement:** Data is contained within the article.

**Acknowledgments:** Natural Interactive Learning Games and Environments Lab (NILE) unit of the Artificial Intelligence and System Engineering Laboratory (AISE) of the Department of Electrical and Computer Engineering (ECE) of the Hellenic Mediterranean University (HMU).

**Conflicts of Interest:** The authors declare no conflict of interest.

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
