# Peer review of "EduARdo—Unity Components for Augmented Reality Environments†"

_information, doi:10.3390/info14040252_

Round 1

Reviewer 1 Report

Presented article with Title ”EduARdo - Unity Components for Augmented Reality Environ- 2 ments” is writing on 15 pages with 10 figures, 1 table and 28 references. The paper is written clearly, but it has a number of shortcomings. The structure is very clear (introduction, related work, matherials and methods, results, discussion and conclusion).

Suggestions:

-          In article is missing a more detailed art of state.

-          In the article lacks a graphic representation of testing work.

-          I recommend the authors to pay more attention to methodology.

-          The authors used general knowledge and the contribution of this article is not clear.

-          What was the application? What area was it focused on? How was AR used?  These questions need to be answered.

-          Which is the novelty of the article, as Unity 3D has an automatic AR module in it and hand tracking is done automatically.

-          Conclusions require redrafting. They are too short and inconsistent.

The quality and information provided in some figures needs correcting. All the specific comments can be followed in revised copy of the manuscript.     

After study I recomend this paper reject.

Author Response

please find authors reply to your comments in the attached file

Reviewer 2 Report

Article provide a framework to assist non-technical person for developing projects in AR devices. I have following suggestion to improve quality of paper:

Results are not presented systematically. Specially, from line 341 to line 362, results are not organized and hard to understand. Kindly present a summary of results in start of paragraph to have reader idea what he will read in upcoming paragraph or show such results as a table. 

Author Response

(The authors gave the same response as above.)

Reviewer 3 Report

The paper describes a system aims on making the use of AR simpler, and the development of the experiences easier. It does this by providing components in the Unity 3D engine.

-          The introduction provides directions for the research, but it lacks underpinning the boundaries and decision made. The authors should explain better why e.g. the focus is on hand interaction, and many other aspects that contribute to an AR experience/environment are not elaborated on.

o   In the introduction some statements are made without clear referencing or logical explanation (e.g. line 50)

-          The paper could benefit by providing insight in the relations and dependencies that various AR components have on each other. By improving one (such as hand tracking), all the consequences on the others should be taken into account.

-          AR is heavily depending on the content being used in. The characteristics of this, and the limitations (and possibilities) it bring to hand tracking (and other interaction techniques) are not considered in the paper. This makes it difficult to know the potential of the solution.

-          The authors should indicate what the requirements and needs of the solution are. The requirements from the stakeholders, technology, environments, etc.. are not mentioned, making it impossible to know where to test and judge the solution on

-          The state-of-art is not very elaborated. Some functions and systems are mentioned, but an explanation on this selection, nor the conclusion or result from it, are well explained.

-          The conditions of use are not stated; it is unclear for what potential experience it is aimed for (entertainment, engineering, training, etc)

-          The authors should add more explanation on why this type of solution is the most appropriate one, and provide a critical reflection on this.

-          Figure 1 is very limited, it needs more explanation or information to be useful.

-          Chapter 3 is very descriptive in the setup of the individual components of the architecture, but no rational, design or development on this solution is provided. While the underpinning of the solution is essential to understand the selected implementation.

-          The experiment is chapter 4 is well performed, but it is unclear for the reader what the purpose of the experiment is. What elements need testing (e.g. the usability of the tool, or the value of the tool). This makes it difficult to understand the value of the outcome of the experiment.

-          The discussion could benefit from adding a critical reflection on the whole solution, and not only on the results from the experiment.  

Author Response

(The authors gave the same response as above.)

Reviewer 4 Report

1. The graphics/graph presented was too small and unable to read (figure 8)

2. What are the criteria of the respondents being selected to interview

Author Response

(The authors gave the same response as above.)

Round 2

Reviewer 1 Report

After reading the edited version of the article, I recommend to publish the article in present form.